# Mineralization of Titanium Surfaces: Biomimetic Implants

**DOI:** 10.3390/ma14112879

**Published:** 2021-05-27

**Authors:** Javier Gil, Jose Maria Manero, Elisa Ruperez, Eugenio Velasco-Ortega, Alvaro Jiménez-Guerra, Iván Ortiz-García, Loreto Monsalve-Guil

**Affiliations:** 1Bioengineering Institute of Technology, International University of Catalonia, 08195-Sant Cugat del Vallés, 08017 Barcelona, Spain; xavier.gil@uic.cat; 2Department of Materials Science and Metallurgical Engineering, Biomaterials, Biomechanics and Tissue Engineering Group (BBT), Polytechnic University of Catalonia (UPC), 08019 Barcelona, Spain; jose.maria.manero@upc.edu (J.M.M.); elisa.ruperez@upc.edu (E.R.); 3Barcelona Research Center in Multiscale Science and Engineering, Polytechnic University of Catalonia (UPC), 08019 Barcelona, Spain; 4Department of Stomatology, Faculty of Dentistry, University of Seville, 41009 Sevilla, Spain; alopajanosas@hotmail.com (A.J.-G.); ivanortizgarcia1000@hotmail.com (I.O.-G.); lomonsalve@hotmail.es (L.M.-G.)

**Keywords:** mineralization, titanium, dental implants, bone bonding, apatite, biomimetic surface

## Abstract

The surface modification by the formation of apatitic compounds, such as hydroxyapatite, improves biological fixation implants at an early stage after implantation. The structure, which is identical to mineral content of human bone, has the potential to be osteoinductive and/or osteoconductive materials. These calcium phosphates provoke the action of the cell signals that interact with the surface after implantation in order to quickly regenerate bone in contact with dental implants with mineral coating. A new generation of calcium phosphate coatings applied on the titanium surfaces of dental implants using laser, plasma-sprayed, laser-ablation, or electrochemical deposition processes produces that response. However, these modifications produce failures and bad responses in long-term behavior. Calcium phosphates films result in heterogeneous degradation due to the lack of crystallinity of the phosphates with a fast dissolution; conversely, the film presents cracks, which produce fractures in the coating. New thermochemical treatments have been developed to obtain biomimetic surfaces with calcium phosphate compounds that overcome the aforementioned problems. Among them, the chemical modification using biomineralization treatments has been extended to other materials, including composites, bioceramics, biopolymers, peptides, organic molecules, and other metallic materials, showing the potential for growing a calcium phosphate layer under biomimetic conditions.

## 1. Introduction

Nowadays, most dental implants are made of commercially pure titanium; generally, grades two and three of titanium are used is due to their osseointegrating properties [1], resistance to corrosion [2], good mechanical properties, and thermic biocompatibility [3,4]. Titanium forms a spontaneous film of titanium oxide, which makes the material more resistant to corrosion. This titanium oxide film allows a low release of titanium ions to the physiological environment when implanted [2]. Many companies apply an acid treatment so that the titanium oxide layer becomes thicker and more stable to improve the resistance to degradation of the dental implant in an aggressive medium, such as the human mouth [5]. 

The roughness helps to ensure that the bone tissue, formed after placement of the dental implant, colonizes the holes and that its growth causes a blockage of the implant with hard tissue. This process is called the mechanical fixation of a dental implant, or secondary fixation. The good osseointegration of dental implants is vital to avoid problems in the fixation of these implants. To achieve this roughness, various methods may be used; however, the most successful is blasting of the titanium with alumina particles of a given size [6,7,8,9,10].

The optimal roughness of the surface for the studied dental implants has been obtained; not only the roughness, but also other physical properties of the surface may favor the speed of osseointegration, such as the wettability, the zeta potential, and the surface energy [11,12,13,14,15,16]. A research group developed a modification in the passivation by chemical agents that improves the surface properties to facilitate selective protein adsorption, which enhances the adhesion of osteoblasts on the surface of the titanium. This modification does not involve creating a bioactive layer on the material, but rather changing the superficial conditions to favor biological fixation and shortening the time for bone growth around the titanium implant [17].

Some implant brands use acid etching to create negative charges on the surface that prevent their migration through immersion in a polar solvent, such as water (Straumann SLA Active), or implant grit blasted with calcium phosphate particles (bioactive) so that residue on the surface helps to generate new bone around the dental implant (Osseotite); others change topographies with a certain porosity of the titanium oxide of the passivation on the surface, exerting high values of passivating potential generated by etching [18,19]. The designed modification consists of etching by NaOH to produce a modification in the titanium oxide (only in the uppermost part of the passivation layer) by forming sodium titanate, which produces a negative charge on the surface. It is well-known that negative charges promote osseointegration once the dental implant is in contact with the physiological liquids [20,21,22,23].

The mechanisms of the bond between a bioactive layer and the bone tissue are complicated as different interactions between the biomaterial and the bone are involved. The interactions occur at the atomic, molecular, and cellular levels. A schematic of the most important mechanisms that occur is shown in Figure 1 [23].

Once the biomaterial is implanted, it comes into contact with the patient’s blood. The surface energy of the implant is an important factor in the chain of biological reactions, since blood can wet in different ways. If the blood wets well, i.e., its behavior is hydrophilic, the surface will interact much more with the blood components than if the surface is hydrophobic. Proteins contained in the blood are adsorbed to the surface. This adsorption process depends on the chemical nature of the surface; the zeta potential; the roughness at the macro, micro, and nano levels; electrical charges; and residual stresses. The ionic exchange between the biomaterial and the organism is the key factor for a material to be bioactive and not biostable [24,25,26,27,28,29]. There are always three mechanisms involved in bioactivity processes: dissolution, precipitation, and ion exchange. Research is needed so that these interactions, including the adsorption of molecules, peptides, or proteins, are selective to those that promote adhesion, proliferation, and differentiation of the desired cells [30,31]. One of the most common cases is the bioactive layers of calcium phosphates, in which the layers have low crystallinity to favor their dissolution, and thus provide the physiological medium with anions and cations needed for the formation of newly formed bone tissue [32,33,34,35,36]. The objective of this review was to provide an overview of the strategies used to accelerate this mineralization.

## 2. Thermo-Chemical Treatment Methodology

The passive layer of titanium oxide (TiO_2_) reacts with sodium hydroxide solution and forms a sodium titanate gel, which is stabilized by the appropriate heat treatment, resulting in the formation of a layer of partially crystalline titanate. Once the sodium titanate layer is formed, it is expected in vitro to be covered with abundant Ti–OH groups through the exchange of Na^+^ ions from the body fluid.

In sodium hydroxide treatment, the titanium oxide that protects the surface is partially dissolved to form an alkaline solution due to corrosive attacks of the hydroxyl groups. If the attacks of the hydroxyl groups continue on the hydrated TiO_2_ surface, negatively charged hydrates are produced on the sample surface. These negatively charged molecules are combined with sodium ions in the aqueous solution, resulting in the formation of a layer of sodium titanate hydrogel.

During heat treatment, this hydrogel is dehydrated and densified, forming a stable layer of sodium titanate, partially crystallized. When the thermo and chemically treated titanium is immersed in body fluid, the amorphous layer quickly releases Na^+^ ions, forming Ti–OH groups on the surface of the metal, which is enhanced as the amount of Na^+^ ions released increases. The formed Ti–OH groups, which are produced by the exchange of Na^+^, are initially combined with the calcium ions from the SBF, forming an amorphous calcium titanate (Figure 2).

The apatite-forming ability of the dental implant is intrinsic within the body when in contact with the physiological fluids of a patient. If desired, this apatite layer can also be obtained in vitro using simulated body fluid. This process is carried out following the guidelines of the international standard ISO 23317 (Implants for surgery-In vitro evaluation for apatite-forming ability of implant materials) [37,38,39,40]. Therefore, the in vivo formation of apatite can be reproduced using simulated body fluid (SBF), with a concentration level similar to that of human blood plasma ions. This means that the bioactivity of the material can be predicted on the surface of the implant by forming apatite when immersed in SBF [40,41,42,43]. Therefore, when the titanium dental implant is treated with NaOH, a reaction occurs in the outermost layer of the titanium oxide that produces sodium titanate. Figure 2 shows the formation of a titanium oxide evolving into a sodium titanate hydrogel. The heat treatment favors the outflow of water for the formation of the sodium titanate. This phase is amorphous and contains mixtures of compounds such as Na_2_Ti_5_O_11_, titanium oxides, and rutile (TiO_2_). The alkaline treatment causes a reaction with the titanium oxide layer and the appearance of hydroxyl groups [36,37,38,39].
TiO_2_ + OH^−^ → HTiO_3_^−^(1)

This reaction is assumed to proceed simultaneously with the following hydration of Ti metal [39,40,41].
Ti + 3OH^−^ → Ti(OH)_3_^+^ + 4e^−^(2)
Ti(OH)_3_^+^ + e^−^ → TiO_2_·H_2_O + 0.5H_2_ (g)(3)
Ti(OH)_3_^+^ + OH^−^ ⇔ Ti(OH)_4_(4)

A hydroxyl reaction to hydrated TiO_2_ produces negatively charged hydrates on the surfaces of the substrates:TiO_2_·nH_2_O + OH^−^ ⇔ HTiO^3−^·nH_2_O(5)

When the hydrogel dehydrates, it densifies as sodium titanate. Upon contact with simulated body fluid, the alkaline titanate rehydrates and sodium ions are released, forming a TiO_2_ hydrogel. The exchange between the alkaline ions and H_3_O^+^ causes an increase in the pH values of the physiological fluid. Similarly, this increase in pH causes an increase in ionic activity, producing apatite [39,42,43]. This is the surface that the dental implant would have just before its implantation. We observed that different sterilization methods do not change either the nature or the topography of the surface of the dental implant, or the optimal roughness with which it was intended to have.
10Ca^2+^ + 6PO_4_^3−^ + 2OH^−^ ⇔ Ca_10_(PO_4_)_6_(OH)_2_(6)

Figure 3 shows the microstructure obtained by scanning electron microscopy, and Figure 4 shows the same technique at a higher magnification.

When performing long-angle X-ray diffraction, we can observe the presence of sodium titanate, and no other element than titanium and the titanium oxide of the passive layer of the dental implant (Figure 5).

When sodium titanate is immersed in a solution simulating human plasma, it shows the appearance of apatite crystals [44,45,46]. This SEM observation is depicted in Figure 6.

These calcium phosphates with cauliflower forms exhibit the same mineral content as human bone. After a short time, the whole surface becomes calcium phosphate, maintaining the roughness of the dental implant (Figure 7).

## 3. In Vitro Assays

Cellular assays were performed using sodium titanate surfaces, and these were compared with other surfaces, showing that the surfaces with the modified passivation presented high adhesion and excellent levels of osteocalcin compared with other surfaces. Figure 8 shows the adhesion rate, and Figure 9 shows the high level of osteocalcin, proving the good cellular response of sodium titanate (Figure 10).

## 4. In Vivo Assays

Klockner’s dental implantations were performed on pigs at different times of osseointegration. In Figure 11, the thermochemical treatment (2S) can be observed, showing that the sodium titanate had a good osseointegration at 4 weeks after implantation. This value is compared with other surface treatments (R: roughened by blasting; E: acid attack; Ctr: control). These values show an acceleration of bone in-growth around the dental implant, which was already significant 2 weeks after implant placement. These implants provided excellent primary fixation, even before the hard tissues started to significantly regenerate. These osseointegration levels evidenced the good short-term in vivo performance of the treated surfaces in comparison with the other conventional implants tested. Osseointegration was accelerated and, consequently, implant failure decreased. Figure 12 indicates the osteoconductive behavior of the thermochemically treated surface two2-step), where it can be observed that the new bone is growing from the surface of the implant [44,45,46].

The histologies show how the implant surface facilitates the formation of bone, as shown in Figure 12 and Figure 13, and where bone is formed from the implant surface with sodium titanate. With this, an osteoinductive effect is achieved that favors the reduction in osseointegration time [47,48,49,50].

## 5. Long-Term Degradation Test

Corrosion tests were performed, and the results of current and polarization resistance were better than those of the implant without performing the modified passivation treatment [2]. Fatigue tests were also performed (cyclic loads that simulated human mastication), and the results met fatigue international standards, surpassing the levels of cycling for high values of exerted mechanical stresses. No statistically significant differences were found with respect to conventional implants, which are used by the same company, SOADCO. Due to their design, the implants present higher resistance values than commercially available implants from other companies [51].

The result of the bond strength between the apatite layer and titanium, obtained by scratch tests, was approximately 570 ± 34 MPa [19,20]; the force of adhesion of the implant bone, measured by pull-out tests at 4 weeks after implantation, was 385 ± 24 N and, at 6 weeks after implantation, was 396 ± 44 N [48].

## 6. Similar Strategies Used by Enterprises in the Dental Implants Sector

The Ospol dental implant uses an oxidized surface with calcium ions present in the electrolyte that remain on the surface after the formation of the passivation layer. A level of 11% calcium on a surface of one to two micrometers in the same layer is obtained. This treatment is similar to that proposed; in this case, it is a modification of the passivation process with the incorporation of calcium in the electrolyte. In our case, it is sodium hydroxide by chemical methods instead of electrochemical.

XPEEDR are Ti implants with deep threads with resorbable blast media surfaces produced by grit blasting, or XPEED surfaces produced by coating of the nanostructure’s calcium. These surfaces are prepared by hydrothermal reactions in a mixed solution containing 2 mM CaO and 0.2 M NaOH at 180 °C for 2 h. All implants are sterilized by gamma irradiation. The mean removal torque of the Ti implants is 28.9 Ncm with the hydrothermal treatment, and 20.5 Ncm for the control. The mean bone index contact (BIC) at 4 weeks after implantation is 53.2% for the Ti implants with rough surfaces, and 73.4% for the Ti with XPEED surfaces. This treatment is similar because the NaOH is the same; in our case, we changed the concentration in order to obtain sodium titanate gel and we did not introduce calcium oxide.

SLActive implants are blasted with alumina particles and then treated with an acid attack that generates an extra roughness on the primary rough surface due to the bombardment of the abrasive. The attack creates a hydrophilic surface with a large number of hydroxy groups. This result is similar to that obtained by treatment of the dental implants manufactured by SOADCO. However, in the case of SLActive, the negative charge of the surface must be maintained; to do so, an immersion in water is performed so that the negative charge is not neutralized by the hydrocarbons in the environment. From the acid attack until the water immersion, the dental implant has a pure nitrogen atmosphere to avoid contact with the atmosphere. Figure 14 shows a micrograph of the surface.

The dental implant of SLA differs from SLActive in the reserve of the negative charge, as the implant loses much of its hydrophilic properties. This implant treatment produces the negative charge of the surface, the mechanism is different, but the surface potential character is similar to our thermo-chemical treatment.

The NanoTite dental implant applies a dual acid etching to roughen, and then dicalcium phosphate nanoparticles are projected to promote activity. In this implant, there is a mechanical deposit by anchoring dicalcium phosphate nanoparticles that promote bone formation. It has been observed in several scientific papers that the bioactive effect is small because there are few particles mechanically retained on the surface of the dental implant. It has been rated on less than 10% of the surface. In addition, there is much controversy about the action of the nanoparticles, since their size is too small to effectively activate the mechanisms of cell transmission.

Some dental implants in which particles or glasses of calcium phosphate are projected appeared on the market, but they failed because the particles were only a little abrasive and crumbled with the impact, and were not retained on the dental implant surface. The proposed solution was to obtain glasses and project them onto the surface of the dental implants, but the sharp edges of the glass particles caused cracks on the surface of the dental implants that may cause premature breakage because of the chewing cycles, which cause fractures by fatigue.

With this implant type, the presence of apatite produces good osseointegration. This implant may be considered a bioactive implant. In our case, we could not consider bioactive implants, but the sodium titanate in contact with the physiological environment produces a human-natural apatite, which is produced by the osteoblast cells. The nucleation and the growth of the bone tissue are natural, and are not induced as nanotite, which has apatite on its surface.

## 7. Other Strategies

One of the most widely used strategies is to create calcium phosphate layers with the same mineral content as bone on titanium surfaces [14], so that cell signaling would activate the migration of osteoblastic cells to rapidly colonize and accelerate the formation of new bone tissue. One of the techniques used has been sprayed titanium plasma [52,53].

This technique causes the projection of calcium phosphate onto titanium from a plasma state at high temperatures (9000 °C). The projection produces a calcium phosphate coating that can be a bioactive element for faster spindle generation. However, this technique, which has been widely used, experiences many problems, as discussed below: [54,55,56,57,58,59,60,61]

The calcium phosphate coating, due to the high solidification rate, cannot create a crystalline apatite layer but rather produces an amorphous calcium phosphate layer. The dissolution of the amorphous layer is much faster than the dissolution of the crystalline hydroxyapatite layer, so that in a short time, the amorphous layer dissolves from the titanium surface and the bone that had colonized the layer is detached from the dental implant together with the amorphous layer.The amorphous layer forms due to the fast solidification rate, which causes the appearance of many cracks due to thermal shock, greatly reducing the mechanical stability of the layer.The lack of chemical bonding between the amorphous calcium phosphate layer and the titanium causes large voids between the metal and the layer, leading to rapid bacterial colonization and facilitating infection.

These are the reasons why implants with bioactive coatings produced by plasma sprayed were withdrawn from the market. In studies of osseointegration of bioactive dental implants by sprayed plasma, a rapid level of osseointegration was observed within a few weeks, but soon after, this osseointegration decreased drastically due to the detachment of the amorphous layer of the dental implant.

Other techniques with slower deposition rates were studied in order to obtain crystalline apatite layers, such as laser ablation, but the cost of the dental implant was high due to the complexity and the amount of energy required for its production.

Zelijka et al. studied coating on TiO_2_, which improves the formation of apatite on it. D3-modified titanium surfaces induce the spontaneous formation of biocompatible bone-like calcium phosphates [62].

## 8. Porous Titanium

One of the problems with metallic materials for hard-tissue replacement is the high elastic modulus, which makes the transfer of mechanical loads to the bone inadequate and causes stress shielding and loss of bone mass. One of the strategies to reduce the elastic modulus is the use of titanium to alloy with beta elements (zirconium, niobium, and molybdenum), which forms beta-titanium-type alloys that can reduce up to half of its stiffness.

Another strategy is to obtain porous titanium materials since, in addition to reducing the elastic modulus, the voids can also be colonized by the bone to improve the anchorage of the implant or the prosthesis with the tissue [63,64,65,66,67,68]. However, the downside of porous materials is their removal in the case of corrosion or infection. Bone colonization of voids leads to complicated surgeries and destruction of bone tissue.

The techniques for obtaining porous materials are diverse: sintering, 3D printing, and obtaining foams by self-combustion [65,66,67,68,69]. Many of the techniques used for obtaining porous materials make it possible to obtain materials with interconnected porosity with the ability to modify the parameters to obtain the desired porosity. The porosity will cause a reduction in both static and cyclic (fatigue) mechanical properties, which should be considered for clinical applications [70,71].

Osteoblastic cells that were on the metallic foams for 7 days are shown in Figure 15. The cell culture of human osteoblastic cells in microspheres have been be observed, with good behavior in cell extension and dorsal activity of the cells [71,72]. No preferential zones for their adhesion have been observed, nor any influence on the studied sizes of the spheres [71,72,73,74,75].

Since the spheres or porous materials obtained are composed of titanium, thermo-chemical treatments have been successfully carried out to obtain the sodium titanate layers that allow the formation of apatite in contact with the physiological medium, accelerating the mineralization process; therefore, the bone colonization will be faster. It was observed that the titanate layer obtained by the treatment is introduced in all the holes and channels of the implant; therefore, the material will have a high osteoconductive-specific surface for bone formation. This finding can be observed in Figure 16. Figure 17 shows a porous titanium material with thermo-chemical treatment in which the colonization and formation of new bone inside the pores of a dental implant can be observed.

## Figures and Tables

**Figure 1 materials-14-02879-f001:**
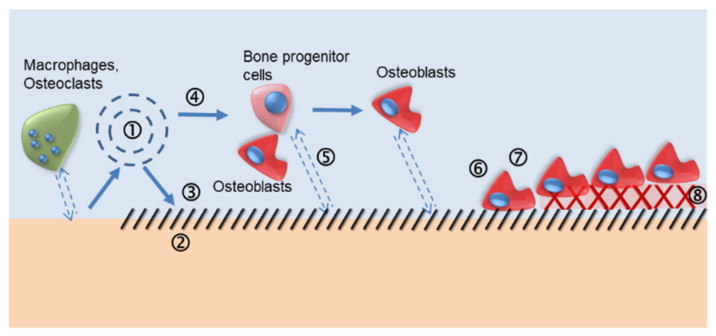
Scheme of bioactive materials’ ability to bond bone tissue. (**1**) Dissolution/precipitation processes: (passive or cell-mediated)/protein adsorption; (**2**) formation of a biological HA layer; (**3**) ionic interactions and new organization at the bioactive surface with the bone tissue interface; (**4**) solution-mediated effects on cellular activity; (**5**) chemotaxis to the biomaterial surface; (**6**) cell adhesion and cell proliferation; (**7**) cell differentiation; and (**8**) formation and growth of the extracellular matrix.

**Figure 2 materials-14-02879-f002:**
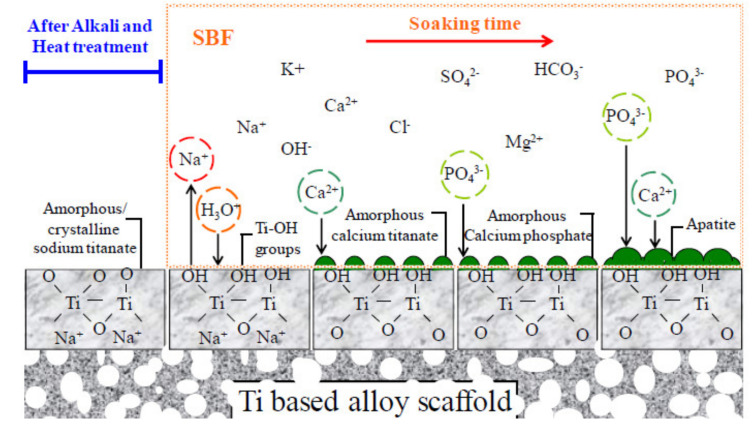
Scheme of the relationship in the process of formation of apatite on the surface structure when immersed in SBF.

**Figure 3 materials-14-02879-f003:**
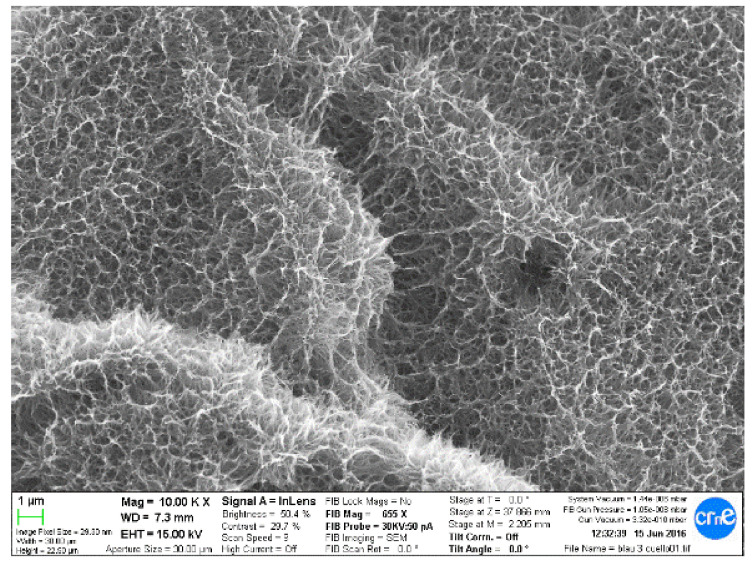
Titanate gel on a titanium surface.

**Figure 4 materials-14-02879-f004:**
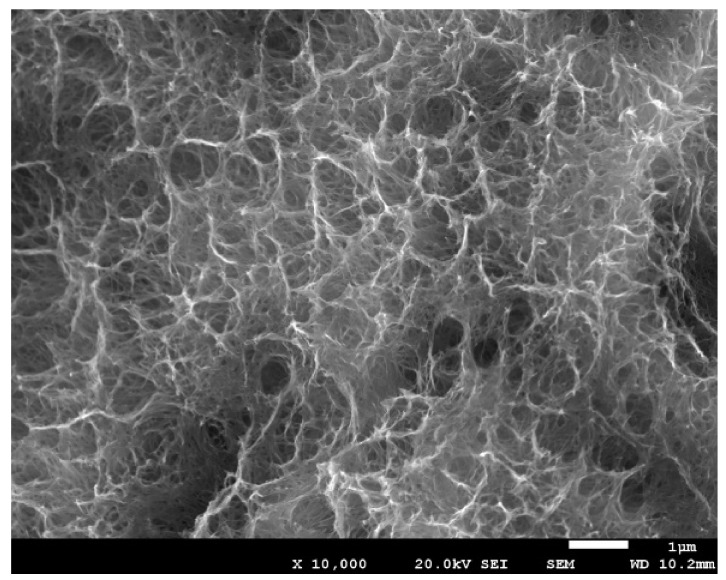
Titanate gel on a titanium surface at a higher magnification.

**Figure 5 materials-14-02879-f005:**
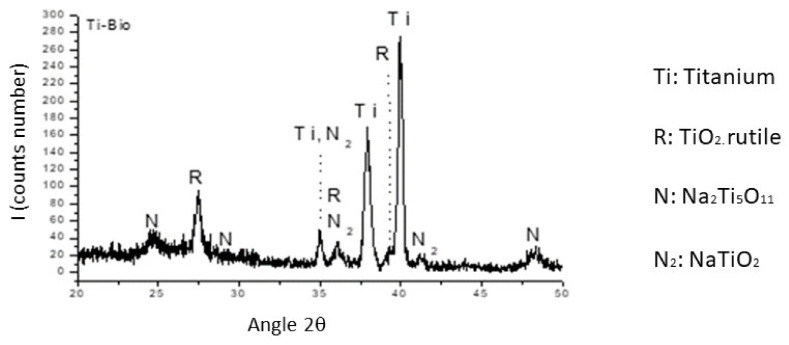
X-ray diffractogram of the titanium surface with apatite.

**Figure 6 materials-14-02879-f006:**
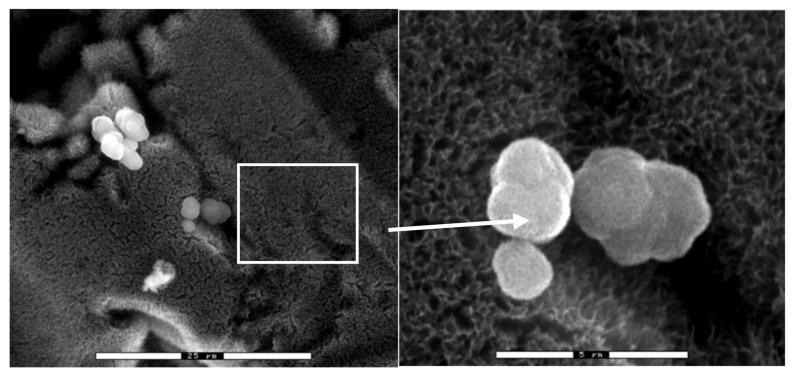
The first apatite nuclei appeared after 3 days.

**Figure 7 materials-14-02879-f007:**
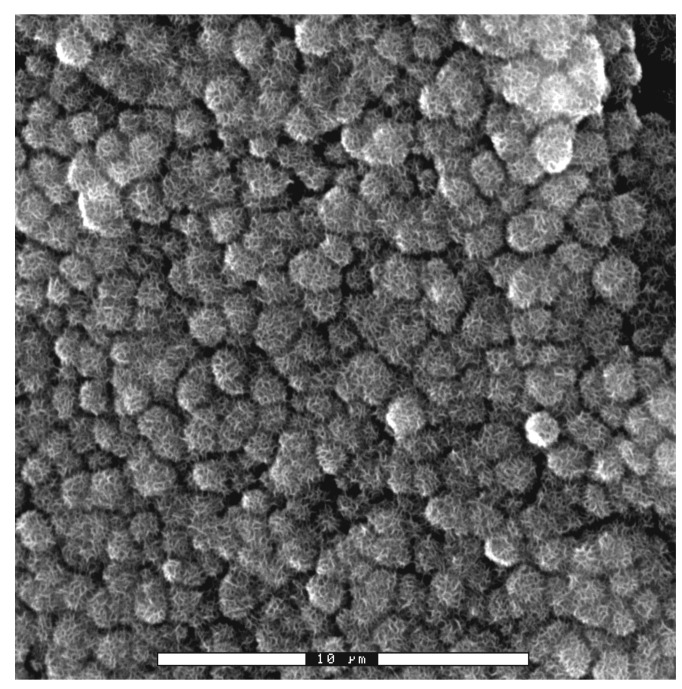
Titanium dental implant surface completely covered with apatite.

**Figure 8 materials-14-02879-f008:**
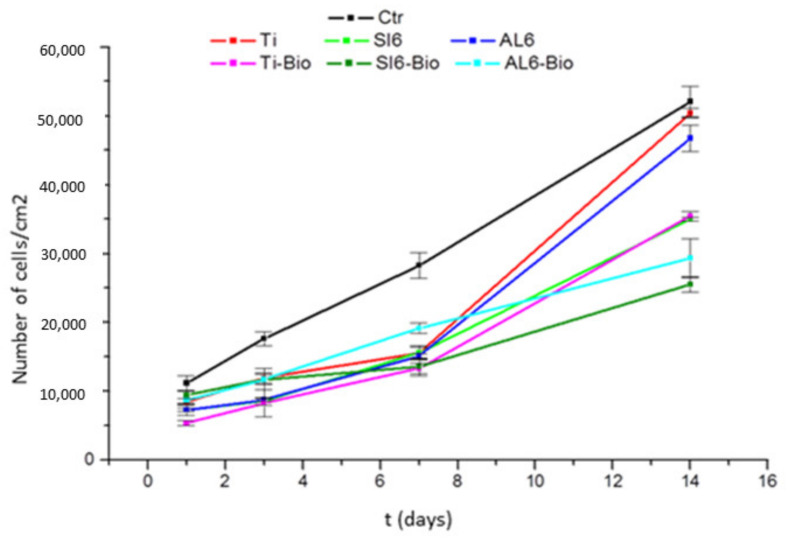
Ctr: control; Ti: titanium cp; Ti-Bio: smooth titanium with sodium titanate; Si6 sandblasting with SiC; Al6 sandblasting with Al_2_O_3_; and Al6-Bio: sandblasting with alumina and sodium titanate.

**Figure 9 materials-14-02879-f009:**
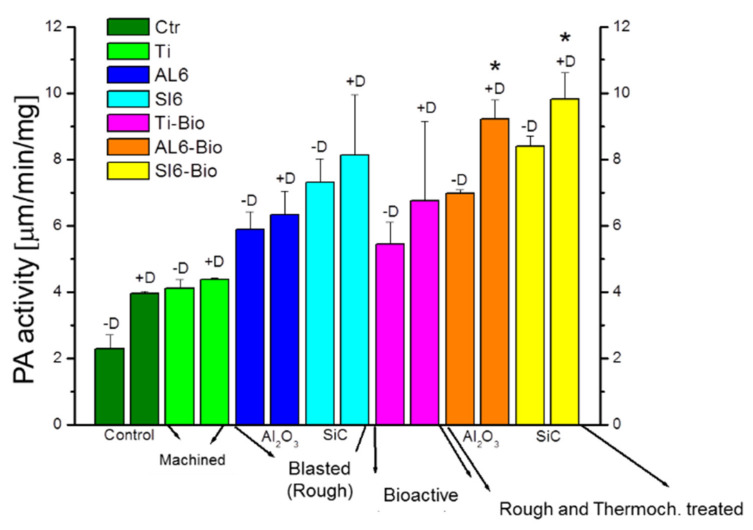
Osteocalcin levels. (* means statistical difference significance).

**Figure 10 materials-14-02879-f010:**
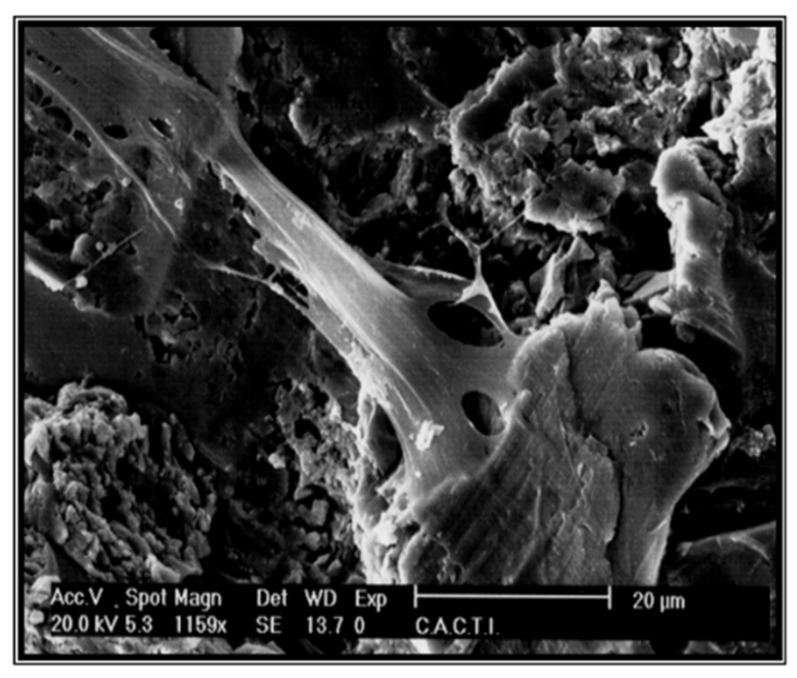
An osteoblast on the surface of sodium titanate with dorsal activity, which is the starting point of new bone formation.

**Figure 11 materials-14-02879-f011:**
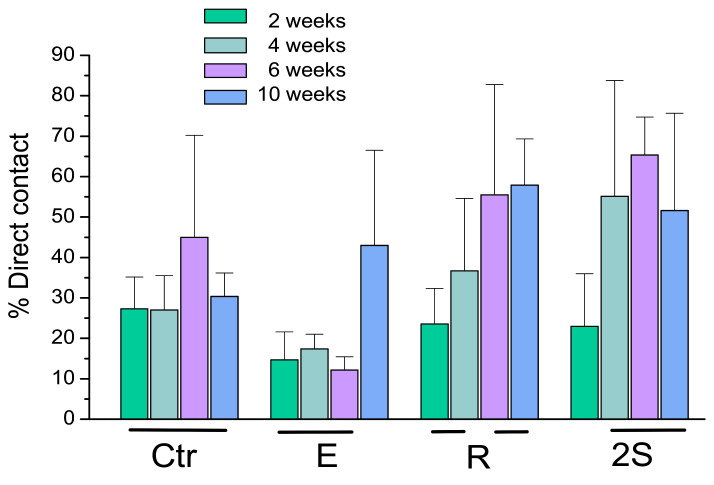
Bone in contact at different times of implantation.

**Figure 12 materials-14-02879-f012:**
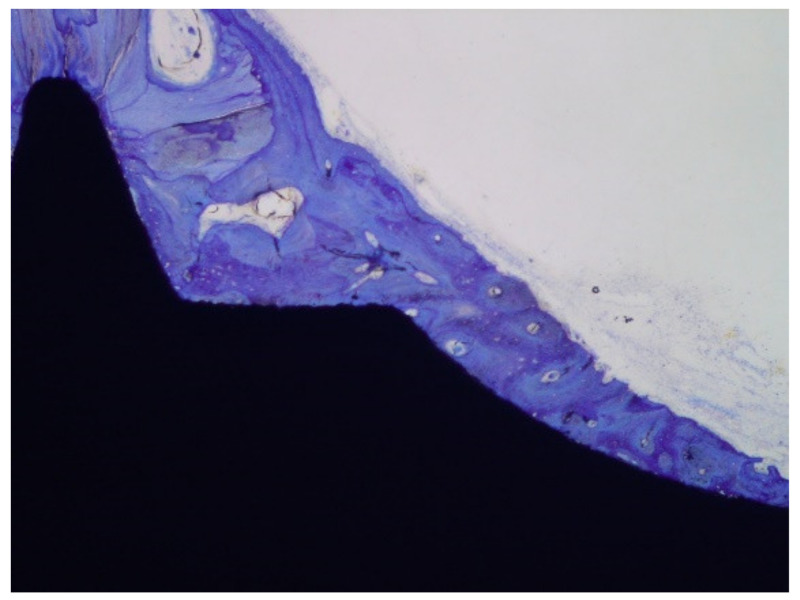
Growth of bone tissue for the dental implant surface.

**Figure 13 materials-14-02879-f013:**
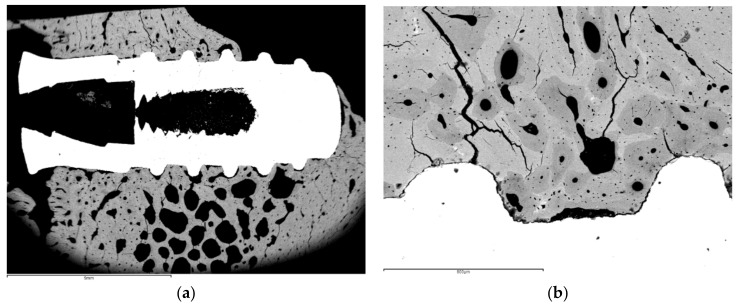
Histology at 2 weeks after implantation. (**a**). Dental implant with an osseointegration of approximately 70% after two weeks of implantation. (**b**). At higher magnification showing the implant-bone tissue interface.

**Figure 14 materials-14-02879-f014:**
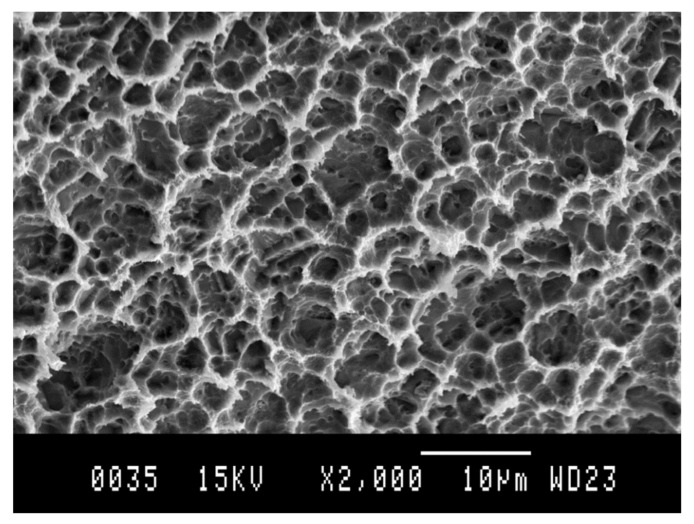
Surface of the SLAactive implant.

**Figure 15 materials-14-02879-f015:**
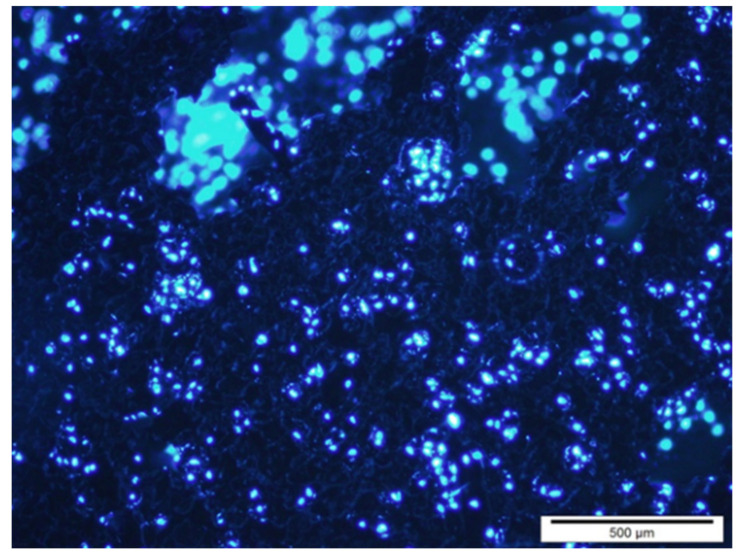
Human osteoblasts on the porous titanium surface at seven days.

**Figure 16 materials-14-02879-f016:**
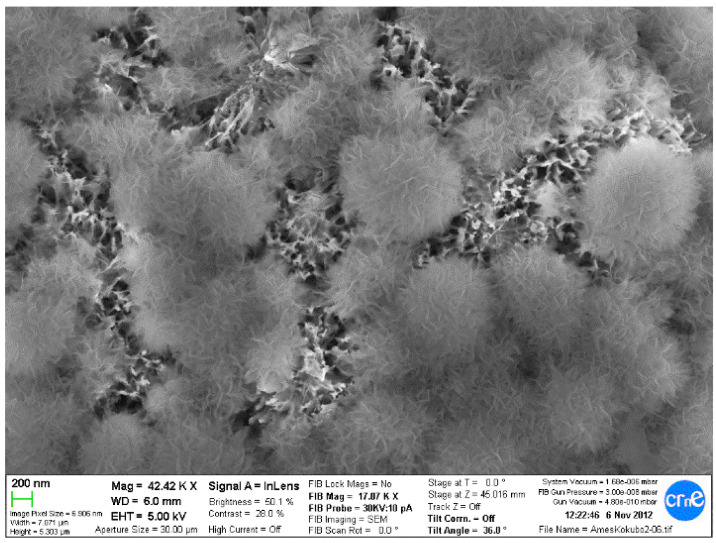
Titanate gel and apatites on the porous titanium surface.

**Figure 17 materials-14-02879-f017:**
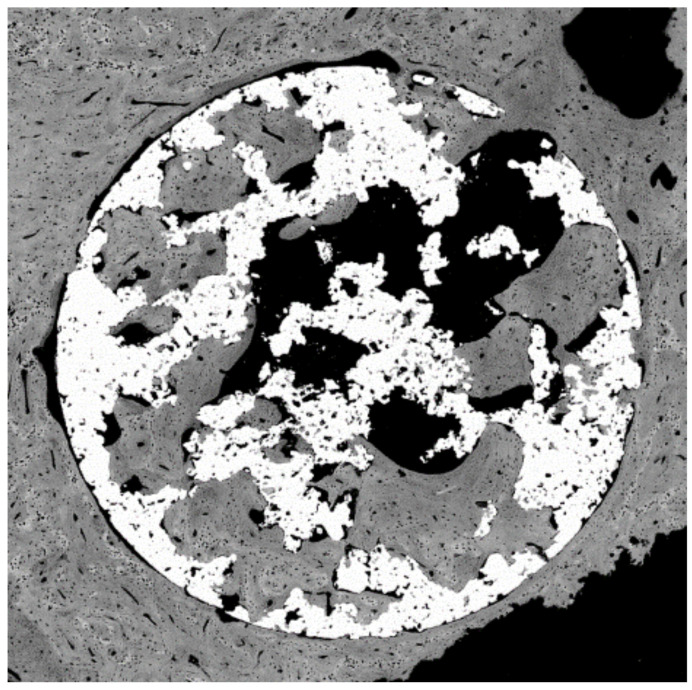
Colonization of bone inside the porous of a bioactive porous dental implant.

## Data Availability

Data sharing not applicable.

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
