# Peer review of "Mineralization of Titanium Surfaces: Biomimetic Implants"

_materials, 2021, doi:10.3390/ma14112879_

Round 1

Reviewer 1 Report

This review describes a biomimetic implant surface modification method that imparts bioactivity to the surface of titanium implants.

The authors recommend surface treatment in which the thermochemically treated titanium implant is immersed in a simulated body fluid as a surface treatment that relieves the shortcomings of existing surface modification methods.

Although this surface treatment method is not novel, in vitro and in vivo experimental results suggest that apatite crystals may precipitate on the titanium surface, leading to early osseointegration.

However, the references in Figures 3-13 and 15-17 are not provided and need to be clarified.

Similarly, L233-273 shows the characteristics of commercial products, but these references must also be provided.

It is also necessary to show the bond strength between this apatite layer and titanium.

Author Response

Reviewer 1

Dear Reviewer,

Thanks for taking the time to review our manuscript and suggest to us to improve our work by providing a lot more detail. We have done so, and we are now submitting a manuscript that not only addresses the points the you specifically raised but also many others that we have considered in order to deliver what we think is a much improved version of our work. This version includes more paragraphs, English grammar revisions in all main sections, new references. Thanks a lot. We are looking forward to your comments.

Sincerely,

Eugenio Velasco

  1. The references have been introduced.
  2. The characteristics of commercial products have been provided with references.ç
  3. The bond strength between the apatite layer and titanium has been introduced at different times of osseointegration.

Reviewer 2 Report

The present manuscript titled "Mineralization of titanium surfaces. Biomimetic implants" is a review of titanium coatings for mineralization enhancement. The manuscript is heavily focused on the groups own work and doesn't describe in depth other comparable techniques.

The manuscript also contains a lot of research images which have not been cited and reprint permissions are not shown if they have been published earlier. The manuscript needs to be restructured to simultaneously compare the different aspects of the coating methods and their effect on mineralization of these implants. The manuscript also has a lot of typological errors. 

Author Response

Reviewer 2

Dear Reviewer,

Thanks for taking the time to review our manuscript and suggest to us to improve our work by providing a lot more detail. We have done so, and we are now submitting a manuscript that not only addresses the points the you specifically raised but also many others that we have considered in order to deliver what we think is a much improved version of our work. This version includes more paragraphs, English grammar revisions in all main sections, new references. Thanks a lot. We are looking forward to your comments.

Sincerely,

Eugenio Velasco

  1. The authors have revised the Figures and the permissions have been given to the Editor. (Figure 11).
  2. The manuscript have been improved in structure and revised the typological errors.

Reviewer 3 Report

Comments on the paper:

The manuscript titled: "Mineralization of titanium surfaces. Biomimetic implants" sounds very interesting, it is well written with the important scientific information.

I have a few minor remarks:

  • Line 22, "...by means laser...“ should be replaced by "...by means of laser... "
  • Line 140, please correct (Na2Ti5O11) and (TiO2) to (Na2Ti5O11) and (TiO2)
  • Line 143, please correct "HTiO3-" to "HTiO3-"
  • Line 146, please correct "Ti(OH)3+" to "Ti(OH)3+"
  • Line 147, please correct "Ti(OH)3+" to "Ti(OH)3+" and "TiO2 H2O" to "TiO2•H2O"
  • Line 148, please correct "Ti(OH)3+" to "Ti(OH)3+"
  • Line 151, please correct "TiO2 nH2O" to "TiO2•nH2O" and "HTiO3- nH2O" to "HTiO3-•nH2O"
  • Line 153, I suggest to insert "hydrogel" in front of "layer" to get "...an alkali titanate hydrogel layer".
  • Line 176, "Studies performed by immersion of“ seems unclear, please reformulate the first part of the sentence.
  • Line 178, Figure 6, I suppose it is obtained also by SEM, please notice that.
  • Line 210 please correct "therochemically" to " thermochemically"
  • Line 250, please reformulate the first part of the sentence. I guess it should be something like "The SLA Active implant is an implant treated by blasting... "

When talking about other strategies, there are studies that indicate that a certain coating on TiO2 improves the formation of apatite on it. More precisely, it has been shown that D3- modified titanium surface induces spontaneous formation of biocompatible bone-like calcium phosphates (CaP). Reference:

Petrović, Željka; Katić, Jozefina; Šarić, Ankica; Despotović, Ines; Matijaković, Nives; Leskovac, Mirela; Kralj, Damir; Petković, Marin; Influence of Biocompatible Coating on Titanium Surface Characteristics // Innovations in Corrosion and Materials Science, 10 (2020), 1; 37-46 doi:10.2174/2352094910999200407095723.

  • Line 336, Figure15, It has been stated, in the manuscript (Line 331),  that the incubation of the cells on the metallic foams  for 1, 7 and 14 days can be observed in Figure 15. In Fig. 15 there is only result at 7 days. What about results at 1 and 14 days?

Author Response

Reviewer 3

Dear Reviewer,

Thanks for taking the time to review our manuscript and suggest to us to improve our work by providing a lot more detail. We have done so, and we are now submitting a manuscript that not only addresses the points the you specifically raised but also many others that we have considered in order to deliver what we think is a much improved version of our work. This version includes more paragraphs, English grammar revisions in all main sections, new references. Thanks a lot. We are looking forward to your comments.

Sincerely,

Eugenio Velasco

  1. All the minor remarks have been corrected according to the reviewer. Lines: 22, 140, 143, 146, 147, 148, 151, 153, 176, 178, 210 and 250.
  2. The authors have introduced the reference suggested by the reviewer about D3 modified titanium surface.
  3. The authors have clarified the Figure 15, as an example of the cell culture.

Round 2

Reviewer 2 Report

Thanks for addressing the concerns.